# A Molecular Dynamics Simulation Study on Enhancement of Mechanical and Tribological Properties of Nitrile—Butadiene Rubber with Varied Contents of Acrylonitrile

**DOI:** 10.3390/polym15183799

**Published:** 2023-09-18

**Authors:** Quan Yuan, Yunlong Li, Shijie Wang, Enqiu He, Bin Yang, Rui Nie

**Affiliations:** 1School of Mechanical Engineering, Shenyang University of Technology, Shenyang 110870, China; m13816818527@163.com (Q.Y.); wang_shijie@sut.edu.cn (S.W.); yangbin0806@163.com (B.Y.); 2School of Chemical Equipment, Shenyang University of Technology, Liaoyang 111003, China; 3Ningbo Institute of Technology, Beihang University, Ningbo 315800, China; nierui1996@126.com

**Keywords:** molecular dynamics simulations, acrylonitrile, NBR, mechanical properties, tribological properties, abrasion rate, RFR, variable importance

## Abstract

The molecular models of nitrile–butadiene rubber (NBR) with varied contents of acrylonitrile (ACN) were developed and investigated to provide an understanding of the enhancement mechanisms of ACN. The investigation was conducted using molecular dynamics (MD) simulations to calculate and predict the mechanical and tribological properties of NBR through the constant strain method and the shearing model. The MD simulation results showed that the mechanical properties of NBR showed an increasing trend until the content of ACN reached 40%. The mechanism to enhance the strength of the rubber by ACN was investigated and analyzed by assessing the binding energy, radius of gyration, mean square displacement, and free volume. The abrasion rate (AR) of NBR was calculated using Fe-NBR-Fe models during the friction processes. The wear results of atomistic simulations indicated that the NBR with 40% ACN content had the best tribological properties due to the synergy among appropriate polarity, rigidity, and chain length of the NBR molecules. In addition, the random forest regression model of predicted AR, based on the dataset of feature parameters extracted by the MD models, was developed to obtain the variable importance for identifying the highly correlated parameters of AR. The torsion–bend–bend energy was obtained and used to successfully predict the AR trend on the new NBR models with other acrylonitrile contents.

## 1. Introduction

The wear phenomenon of polymer materials has received considerable attention as it directly affects the reliability, stability, and safety of materials in industrial applications [1,2,3,4,5,6]. As one of the most attractive matrixes in polymers, nitrile–butadiene rubber (NBR) has exhibited outstanding wear resistance, chemical resistance, and a wide range of operating environments [7]. On the other hand, NBR is the most widely used polar rubber in petroleum industries and mechanical seals due to its moderate cost, good processability, and easy modification [8,9]. Therefore, the mechanical and tribological properties of NBR have been investigated by many experimental researchers [10,11,12,13,14]. The most common method for improving wear resistance of NBR is based on the increasing acrylonitrile (ACN) content to raise the polarity and rigidity of polymer, thereby enhancing material mechanical and tribological properties [15,16]. It has to be noted that although the above experimental studies have acknowledged the best performance of 40% ACN content of NBR, to the authors’ knowledge, few theoretical or molecular simulation studies have been identified. In order to guide the NBR modification in tribology applications, studies are also significant and useful in determining the enhancement mechanism of NBR as the optimizing direction from an atomic level.

As an effective tool for studying materials [16,17,18,19,20], molecular dynamics (MD) simulations can calculate the properties of the polymer model at the macro level and provide details of molecular interactions to explain the mechanisms of action on the atomic scale. Several themes that emerged from investigators have reported on the physical properties of polymer composites formed by MD simulations, such as the glass-transition temperature, mechanical and tribological properties, tribo-chemical properties, thermal properties, and damping properties [4,21,22,23,24,25]. Data from Li [26,27] suggested that the tribological properties of polymer composites under the different variables of normal loading and sliding velocity could be enhanced by the introduction of carbon nanotubes based on MD simulations, where physical interpretations were provided from microscopic information. It can be indicated from the above studies and other research [28,29,30] that the destruction of materials is more from the transfer of external kinetic energy into the potential energy of the polymer by the microscopic perspective of simulation analysis. This means that improving the interaction energy of molecular chains of a polymer is more important for reducing the potential energy of the polymer.

In the MD simulations, the polymer properties were directly or indirectly determined according to the system energy by parameterizing the structural features of polymers [31]. The potential energy of a system can be expressed as a sum of valence (or bond), non-bond interactions, and cross term: *E_total_* = *E_valence_* + *E_non-bond_* + *E_crossterm_* [32]. In addition, these energies can be further subdivided into a dozen specific parameters, such as the bond energy, the angle energy, the Van der Waals energy, and the bend–torsion–bend energy, etc. The importance of these specific parameters on influencing the properties of polymers was often neglected in previous MD theoretical studies due to the large quantity of parameters. These energies are likely to be key correlation factors in the study and prediction of material properties [33,34]. The random forest technique is able to simultaneously analyze hundreds of feature variables [35,36,37,38,39] and produce the ranking of feature importance based on the predictive model of machine learning. A study by Betina [40] proposed a basic method that obtained information on the most important variables from the random forest model, and it was shown to be effective for the distinction of the variables that best contributed to the discrimination of oils.

Considering all the aforementioned information, the mechanical and tribological properties of NBR were calculated and analyzed by MD simulations. Thereinto, according to the universal components of NBR [41], the NBR models considered ACN contents of 20% to 40% with incremental contents of 10%. To determine the atomic mechanisms of enhancements of ACN, the binding energy, radius of gyration, mean square displacement (MSD), and free volume were calculated from the NBR system. In addition, the results and energy values were collected from the MD model for identifying the high correlation parameters of wear performance in different ACN contents of NBR.

## 2. Models and Methods

### 2.1. MD Simulations

MD simulations were performed with Materials Studio 8.0 software. Molecular modeling and dynamic calculation of 4 NBR models were carried out using Amorphous Cell and Forcite Modules. The COMPASS force field [32], which is an ab initio force field that can accurately predict the structural characteristics of a wide range of molecules and polymers, was implemented. These models were constructed as follows: A single chain consisting of 20 repeat units was built with acrylonitrile ($C_{3}H_{3}N$) and butadiene (${CH}_{2}=CH-CH={CH}_{2}$) in 1:4, 3:7, 2:3, and 1:1 allocation ratios [10,11,12,13,14]. The chain length was measured from the atomic distances between the two ends [18]. Each chain was individually packed into an empty cell with a size of 4.5 × 3.0 × 3.0 nm^3^ and periodic boundary conditions until the cell density reached 0.97 g/cm^3^ [22]. The molecules in a cell were built with a Monte Carlo style by minimizing close contacts among atoms whilst keeping a realistic distribution of torsion angles. The information of the NBR models is listed in Table 1.

Since the internal energy was too high in the original NBR models, geometry optimization was applied, and the conjugate gradient algorithm [42] was adopted with an energy convergence of 10^−5^ kcal/mol. The equilibrated structure of the models was obtained further using the NVT (constant volume, constant temperature) and NPT (constant pressure, constant temperature) ensemble. A 600 ps NVT was conducted under a temperature of 298 K. On the last snapshot of the NVT trajectory, an NPT ensemble was carried out for 600 ps at a temperature and pressure of 298 K and 101 KPa. These ensembles were simulated using the Andersen thermostat [43] and Berendsen barostat [44] with time steps of 1 fs. Meanwhile, the Ewald and atom-based summation method was applied to calculate the electrostatic and van der Waals (vdW) interactions with an accuracy of 0.001 kcal/mol and a cutoff distance of 1.25 nm, respectively.

To investigate the mechanical properties of the models, the constant strain method was applied to calculate the Young’s modulus (*E*), bulk modulus (*B*), and shear modulus (*G*). Next, another 200 ps NVT was calculated to obtain the trajectory, in which 4 strains and a maximum strain (*ε*) amplitude of 0.003 were applied to compute the moduli. The elastic constant matrix was estimated by a series of finite difference approximations. In each Cartesian direction, *E = σ/ε* was applied, according to the virial stress definition [45]. By combining the Voigt [46] and Reuss [47] methods with Hill’s definition [48], *B* and *G* were estimated by the 6 × 6 stiffness and compliance matrices accordingly.

To predict the tribological properties of NBR, Fe-NBR-Fe models were proposed and equilibrated by geometry optimization, and the abrasion rate (AR) of NBR was calculated. Fe atom layers with dimensions 4.5 × 3.0 × 0.86 nm^3^ were developed and used as the top and bottom slip planes (the intermediate layer from the equilibrated structure of the models). A 5-cycle annealing simulation with an initial temperature of 150 K and a mid-cycle temperature of 350 K was then conducted by a 200 ps NVT. Finally, the confined shear was realized by applying a sliding speed of 0.1 Å/ps for 500 ps on the upper Fe layer (up-Fe-layer). After the end of the shearing process, AR *= M_leave_/M_total_* was applied, where *M_leave_* and *M_total_* represent the number of atoms that leave the NBR matrices during the friction process and the total number of atoms in the original NBR matrices, respectively. By simulating the friction process, F_N_ and F_f_ (the normal and friction forces on the up-Fe-layer) were obtained, and the AR was measured at the last snapshot of the shearing trajectory.

### 2.2. Datasets

In order to identify the high correlation parameters of AR, the forest model of predicting AR needed to be constructed by feature variables of NBR. In this study, the feature parameters of NBR as the dataset variables, including 27 input variables and AR output variable, were collected from the MD model. The dataset consisted of 5 parameter modules: structural parameters, mechanical properties, environmental variables, friction parameters, and energy parameters. These parameters were derived from the Fe-NBR-Fe model before the shear process, except the AR and mechanical properties, as clearly demonstrated in Figure 1 and Table 2.

Thereinto, the specific parameters of energies were obtained from the empirical potential function [31,49,50] of the force field and are summarized by the expressions below.
(1)Hamiltonian=Total kinetic energy+Total potential energy
(2)$$E_{totalpotential}=E_{valence}+E_{crossterm}+E_{non-bond}$$
(3)$$E_{valence}=E_{bond}+E_{angle}+E_{torsion}+E_{out-of-plane}+E_{Urey-Bradley}$$
(4)$$\begin{array}{c} E_{crossterm}=E_{Bend-Bend}+E_{Bend-Torsion-Bend}+E_{Torsion-Bend-Bend\;}\\ +E_{Torsion-Stretch}+E_{Stretch-Bend-Stretch}+E_{Stretch-Torsion-Stretch\;}+E_{Stretch-Stretch} \end{array}$$
(5)$$E_{non-bond}=E_{van\;der\;Waals}+E_{Coulomb}+E_{hydrogen\;bond}$$

To enrich the dataset, the normal loading and sliding velocity were selected as the feature parameters of the environmental variables to additionally simulate the friction process on each NBR model. The normal loadings were divided into 11 levels, ranging from 0 to 500 kcal/mol/Å, at a sliding speed of 1.1 Å/ps. The sliding velocities were classified into 2 levels with gradients of 0.01 Å/ps (0.03–0.09 Å) and 0.1 Å/ps (0.1–1.0 Å). In summary, 112 samples of the friction process were obtained for forming the original dataset.

It is worth noting that the energy parameters were equivalently designed as the structure descriptors of polymer through MD simulations in this study. The structural parameters of the dataset were also added to explore whether the structural descriptors can be further simplified by atomic information of models.

### 2.3. Identifying the Correlation Parameters of AR

Random forest is one of the most popular supervised machine learning approaches, and it can be applied to solve a wide range of prediction problems [37] with a high accuracy compared with other currently used algorithms. Aside from being able to deal with high-dimensional variables without deleting variables, the method has been highly recognized for its inherent procedure of producing the variable importance (VI) [38,39]. From the VI of the prediction model, the importance variables of the AR effect can be initially screened for identifying the correlation parameters. Hence, the random forest regression (RFR) model of predicted AR was developed to obtain the VI of feature parameters. According to the original dataset, different decision trees were constructed to develop the RFR model, as shown schematically in Figure 2.

First, a new dataset for each tree was established using the bootstrap strategy. Next, the new dataset was split into a training dataset (approximately 70%, in-bag) and a test dataset (approximately 30%, out-of-bag). A 4-fold cross validation [51] was then applied to avoid overfitting in the bag step. The training dataset was divided into 3 training subsets and a validation subset. To separate the feature variables from the 3 training subsets, predictive variables were identified within the random selection by the mean squared error (MSE) criterion and were used to set the nodes during regression tree construction [52]. Additionally, hyperparameters of the separation were optimized by the validation subset. The final model of the decision tree was formed by 4 separate results.

The whole process of building the RFR model involved iteratively repeating the growth process of one tree until the total number of trees reached 100 (*n_tree_* = 100). The rules and judgment values from all decision trees were then integrated and averaged. According to the contribution proportions of the feature variables, the VI was calculated.

The correlation variable of AR from feature parameters was identified by analyzing the top parameters in VI. The specific analysis is detailed in the RFR results of Section 3.3.

## 3. Results and Discussions

### 3.1. Mechanical Properties

The mechanical properties of NBR were investigated by varying the ACN content. According to the 2.1 MD simulations, the mechanical modulus of NBR was simulated 3~5 times for each model and reported the property values from the average of 3 times simulations. As illustrated in Figure 3 and Table 3, *E*, *B*, and *G* values of these models of 3.05 to 3.80, 2.95 to 3.39, and 1.25 to 1.42 GPa, respectively, were obtained with increasing values of ACN. Compared with the monograph [32], the research also found that increasing the ACN content reinforced the mechanical properties attributed to the polarity and rigidity of NBR. The polarity atomic nitrogen with an increase of the ACN group enhanced rubber polarity to increase the interactions of NBR molecular chains. According to the conformation of polymer chains, the flexibility of NBR molecules with an increase in ACN were reduced, owing to reduced double bonds and increased side groups. Meanwhile, the reduced chain length and increased molecular rigidity caused a decrease of entanglement between the NBR molecular chains.

In this study, these moduli approximately showed an increasing trend until the content of ACN reached 40%. In addition to the Bulk modulus, the Young’s and shear moduli of ACN 50 were only slightly increased and decreased compared to ACN 40. Overall, ACN enhanced NBR mechanical properties and showed a turning point at 40% content. A possible explanation for this is that 40% ACN represents a turning point for the dominant role of the reinforcing effect in NBR. The increasing polarity and rigidity of NBR with an increased ACN content enhanced the interactions of NBR molecular chains until the strong rigidity of NBR limited the movement and entanglement of chains after this point.

In order to determine the atomic mechanisms responsible for the enhancement in the mechanical properties, the interaction of NBR molecular chains was calculated during the NPT equilibrium processes. A complete molecular chain from each model is marked as an NBRL in Figure 4 (shown as a red chain in Figure 1 in some cells, too). The binding energy (*U_bind_*) between the NBRL and the NBR matrix was calculated as shown in Figure 4 using Equation (6):(6)$$U_{bind}=-U_{inter}=-\left(U_{total}-U_{NBRL}-U_{NBR}\right)$$
where *U_total_* represents the total energy of the NBR, *U_NBRL_* identifies the energy of the NBRL, and *U_NBR_* represents the energy of the NBR, except for the NBRL. In order to reduce the errors of different NBRL, the average value of the binding energy of NBRL was calculated and repeated 10 times to further average as shown in Figure 4. The *U_bind_* increased from 195.70 to 206.52 kcal/mol as the ACN content increased, indicating that having more ACN enhanced the interaction of NBR molecular chains by the increase in polarity and thus improved the mechanical properties of NBR. It was also observed from the structure of the NBR chain that the degree of coiling of molecules with an increase in ACN was stronger. The cause of this phenomenon can be explained that the double bond number of models was decreasing to reduce the bond angle value of the NBR backbone, and the polarity of the models was increasing to enhance intramolecular interaction force.

In the NPT equilibrium processes, the radius of gyration of NBR molecular chains of models was monitored as a parameter to assess the flexibility of NBR and to show the influence of ACN contents on the movement of NBR molecular chains. As shown in Figure 5, the radius of gyration of NBR molecular chains with increasing values of ACN was reduced, indicating that NBR molecular chains with an increase in ACN had a smaller radius of gyration, lower flexibility, and more rigidity. This comparison showed that the increasing of the ACN group restricted the flexibility and enhanced the rigidity of NBR molecular chains, thereby improving the shear resistance and further the mechanical properties of NBR. Meanwhile, the standard deviation of the radius of gyration of models by an increased ACN content declined, which showed that the ACN group limited the range of motion of NBR molecular chains during the NPT equilibrium process. Especially, the standard deviation of ACN50 value was 0.03 Å, showing decreases of 30%, 57%, and 40% compared with others, respectively. The extremely high rigidity of the ACN50 model limited the entanglement of NBR molecular chains, thus little improving the mechanical properties of ACN50 than ACN40 of NBR.

To verify the above hypothesis of a 40% turning point, the MSD of the NBR molecular chains, which plays a vital role in determining the movement of the polymer chains [53], was calculated during the NPT equilibrium process and is recorded in Figure 6. The average MSD values of models with increases in ACN were 3.51, 3.01, 2.66, and 2.05 Å^2^, showing decreases of 14.26%, 24.30%, and 41.61% compared with the ACN20 model, respectively. In comparison with the other models, the flatter slope of the ACN50 curve indicates that the motion of NBR molecular chains was greatly limited due to its excessive polarity and rigidity, and thus the mechanical properties of ACN50 barely improved relative to those of ACN40. This restriction largely dampened the entanglement between molecules, affecting the enhancement of strength during the equilibrium processes.

In addition, to further assess the interactions and movement of NBR molecular chains, the free volume was measured based on the equilibrated structures of models, as recorded in Figure 7. The field volume was calculated as *Field Volume = Enclosed Volume + Free Volume*. The Connolly surface was calculated when a probe molecule with a radius (R_p_) of 1.0 Å rolled over the vdW surface, and the enclosed volume and free volume were determined as the volumes on the side of the Connolly surface with and without atoms, respectively. The trends of the field volume (the blue and gray bar) and enclosed volume (the gray bar) slightly decreased as the ACN content increased, which indicates that the interactions and movement of NBR molecular chains were enhanced and limited with more branches of the ACN group, leading to NBR molecular chains with a higher polarity and stronger rigidity. These results agree with the above-mentioned analysis of mechanical properties.

It is interesting to mention that ACN30 had the peak value for *B*, *U_bind_*, and free volume for all models. The standard deviation of the radius of gyration on ACN30 (0.07 Å) was more than on ACN20 (0.06 Å), which meant the range of motion of NBR molecular chains was larger on ACN30, in agreement with the result of free volume. Under the same increment on polarity of NBR, the distance of intermolecular interactions increased, which led to a decrease of *B* and *U_bind_*, rather than a rise similar to others. Compared with the *E* and *G*, the relative position of intermolecular could be considered as a key factor on the mechanism of *B*. The polarity and rigidity of NBR were critical factors to determine the performance on the *E* and *G*.

In addition, there are many different initial polymer configurations in the same ACN content due to the complex polymer configurations. New models of 30% ACN content were additionally built from other monomer arrangements to validate the ACN30 finding. From Table 4, the 1211 (the original ACN30 model) and 2121 items contained 2 ACN consecutive units, and the 2130 and 3111 items contained 3 ACN consecutive units. According to the above analytical methods, the *U_bind_*, and free volume of these models were calculated and shown in Table 4 and Figure 8.

As shown in Figure 8, the *U_bind_* values of 30% ACN content models were lower than the others, and the free volume of these models were larger than the others. The results showed that the peak value of ACN30 was not by accident, and the analysis of key factors of the mechanism was meaningful. Moreover, ACN30 models starting from the different structural configurations could generate slightly different values as shown in Figure 8, but these results did not affect the analysis of other systems in the study.

The next section further analyses the tribological properties of NBR by observing the friction process of models.

### 3.2. Tribological Properties

To investigate the performance of the friction and wear of the NBR matrix with an increase in the ACN content, snapshots of NBR models during the shearing process were observed and are recorded in Figure 9. The phenomenon of adsorption occurred at the interface between the Fe and NBR matrices owing to the significance of the attractive vdW forces and leading to the wear behavior. AR values of 36.76%, 30.94%, 20.90%, and 24.33% were obtained for the models as the ACN content increased. These values represent decreases of approximately 16%, 43%, and 34% as compared with ACN20 (36.76%). It can be observed from Figure 9 that with an increase in the ACN content, the NBR molecular chains were surrounded by more atoms from the NBR matrices, and the relative slippage appeared more clearly between the friction interfaces. As also indicated, the tribological properties of NBR can be enhanced due to the increasing polarity and rigidity with an increase in the ACN content, leading to greater interactions, greater stiffness, and adsorption reductions of NBR until an ACN content of 40% is reached.

To further verify the above discussions, the depth variations in the up-Fe-layer of the models were recorded to observe the deformation of the NBR when the external force of the up-Fe-layer was removed. The rising amplitudes of the up-Fe-layer with increases in the ACN content were 0.71, 0.25, 0.22, and 0.34 Å, respectively. Compared with the value of ACN20, the others were significantly lower, indicating that greater interactions and stiffness were achieved with an increase in the ACN content. The smallest value of ACN40 also represented the best tribological property of NBR, owing to stronger interactions and tighter entanglement, as shown for ACN40. The best result was in good agreement with the AR of ACN40, which was also consistent with the 40% turning point of the mechanical properties of NBR.

Furthermore, the properties of the NBR matrix with ACN50 declined, which could have been due to the state of NBR molecular chains of ACN50, as shown in Figure 9. The NBR molecular chains with the shortest length were subjected to limited movement and less entanglement, causing a smaller interaction area between the molecule contacts. Thus, the AR value of ACN50 increased during the shearing process, and the tribological properties of NBR had no improvement with increases in the ACN content. This meant that the mechanism to enhance NBR by the acrylonitrile was the positive effect of the polarity combined with the negative effect of the rigidity to affect the tribological properties of NBR after 40% ACN content.

Overall, the results of the analysis are consistent with experimental observations [9,52]. The results prove that the variable parameters of these models obtained by performing MD simulations are reliable, and they will become fundamental to the RFR dataset coming from Section 2.2.

### 3.3. RFR Model Results

To avoid the randomness of the RFR model, the datasets were trained multiple times by the RFR algorithm and an importance ranking of the variables was produced. The feature variables of F_N_ and F_f_ were selected as the key factor affecting AR to screen RFR models. Table 5 lists eight RFR models and shows the VI names and values (the proportion of VI) from the top five for obtaining the highly correlated variables of AR from dataset. Not surprisingly, apart from the F_N_ and F_f_, the temperature was also related to AR. The bend–torsion–bend (BTB), torsion–bend–bend (TBB), torsion–stretch (TS), and stretch–torsion–stretch (STS) energies as the structure descriptors are also shown in Table 5. Compared with the AR value of 112 samples, the value of TBB energy was highly correlated with the trend of AR value.

To analyze all parameters of the dataset, the VI of the RFR Model 2 was determined and is shown in Figure 10, which was distinguished according to the parameter modules (from Table 2). The energy parameters, environmental variables, and friction variables occupied 98% of all modules in percentages of 73%, 13.8%, and 11.2%, respectively. As the structure descriptors, the energy parameters mainly contributed to the prediction ability because they have greater influences than external conditions on polymers [54].

In addition, for example, the ACN20 model at a speed of 0.1 Å/ps, the total potential, valence, cross-terms and non-bond energy values were −204,951.00, −1029.05, −672.13, and −202,235.40 kcal/mol, respectively. The total cross-terms energy (59.70% of VI) contributed less energy value than others, leading to its significance often being ignored in previous studies of the energy effects using MD simulations. The cross-terms energy demonstrates the importance of accounting for structure, energy, and dynamics, as shown by JON R. MAPLE [55], and was further used to analysis and predict the AR of NBR in this work.

Figure 10 also summarizes the top 10 VI values and shows that the BTB and TBB energies were superior to others. To explore the effects of these parameters on the wear performance of NBR, the BTB and TBB energy values were obtained from the end of each equilibrium process, including the structure of the packing after geometry optimization, i.e., the structure of the NVT ensemble, the structure of the NPT ensemble, and the structure of the annealing simulation. It was found that the BTB energies obtained from the structures of the packing and annealing processes were highly correlated with the ACN contents of the models, as can be seen in Figure 11A.

The most striking result emerged between the TBB energy and the AR of NBR, as shown in Figure 11B. The same trend was observed from the structure of the packing and annealing processes together with the AR. This may mean that the AR trend of NBR could be estimated by contrasting the TBB energy of NBR at the early stage of the model’s construction. A bold hypothesis was proposed that the value of TBB energy, from the NBR model of the packing after the geometry optimization, could directly predict AR trends for NBR model of other ACN contents. This conjecture was tested and verified by Section 3.4.

It is also worth clarifying that the TBB energy of ACN40 compared with other models had the minimum value at the end structure of any process, which is in agreement with the turning point of a 40% ACN content. This result could be explained by MD simulations showing that the lower TBB energy of NBR could prevent the relative movement of molecules, thereby maintaining the stability of the mechanical properties.

### 3.4. Test and Verification of the Hypothesis

In order to verify the agreement in the trend between the TBB energy and AR of models, new models were packed using the other ACN contents, including proportions of 15%, 25%, 35%, 45%, and 55%. The TBB energy of all models was summarized from the packing structure after geometry optimization, as shown by the pink line in Figure 12. ACN25, ACN15, and ACN55 had, respectively, the maximum values of TBB energy and extrapolated data points for all models, as shown in Figure 12. Therefore, they were selected as the validation models simulated by the MD simulations presented in Section 2.1.

As previously predicted, the AR values of validation models agreed with the trend of the TBB energy, as shown the green bars in Figure 12. This meant that the TBB energy was found as the highly correlated parameter of AR by screening VI from RFR model, which is meaningful and used to roughly predict the AR trends. From the 15 to 25% ACN models’ snapshots, the AR values were increased due to the more atomic adsorption around the interface between metal and polymer materials. This suggests that the increasing polarity in 15–25% ACN contributes more to the vdW force at the interface of the Fe and NBR matrices. In the ACN55 model, NBR emerged with a brittle fracture leading to the higher AR value, owing to the decreases in molecular chain length and an increase in rigidity, as discussed in Section 3.2. These results show that TBB energy can be used as a parameter to predict the trend of AR and the wear performance of NBR with different ratios.

Combined with Figure 9, the lower AR values of NBR had the following characteristics: the stronger interaction force between molecular chains and the lower vdW force at the interface of the Fe and polymer matrices. In particular, the wear results of ACN25 and ACN55 showed that the up-Fe-layer adsorbed more matrices, leading to AR values that were more than others. Therefore, the modified NBR can be considered to enhance the interaction force of molecules and reduce the vdW force between the matrix and Fe layer.

In this study, although no more NBR models were validated, the method of identifying VI by introducing the RFR model still can be used to assist MD analysis in obtaining additional information quickly and efficiently. It provides new solutions for analyzing the mechanism, predicting the performance, and guiding the design in the development of new polymer materials. More precise prediction and analysis using important variables will be deeply investigated in future.

It is worth noting that the regression analysis of RFR model, for treating the use and prioritization of multivariate parameters and their impact on quantitative prediction performance, is widely applicable and not limited to the dimension and variety of data. Developing multiple RFR models is suggested to avoid the randomness of model generation, and selecting the best one for further the analysis and application. In addition, classification problems can be also investigated using the random forest classification algorithm.

## 4. Conclusions

In this research, the effect on the mechanical and tribological properties of NBR with varied ACN contents was studied using MD simulations. The mechanical and tribological properties with increasing ACN were enhanced until the content of ACN reached 40%, and these properties of ACN50 system were unimproved and descended, respectively. The entanglement and interactions of ACN50 molecules were not strengthened by increased polarity as compared to those of ACN40, owning to the stronger rigidity and shorter chain length of ACN50 molecules. In addition, the VI of RFR models is effective for obtaining the highly correlated parameter of AR, getting the TBB energy to predict the AR trend for new models. Meanwhile, the wear forms of new models showed that the vdW force at the interface of the Fe and NBR matrices was a key factor for affecting tribological properties of NBR. In summary, NBR modification is able to enhance the tribological property by increasing the polarity and decreasing the vdW force of the interface, while focusing on the rigidity and chain length in the polymer molecules.

## Figures and Tables

**Figure 1 polymers-15-03799-f001:**
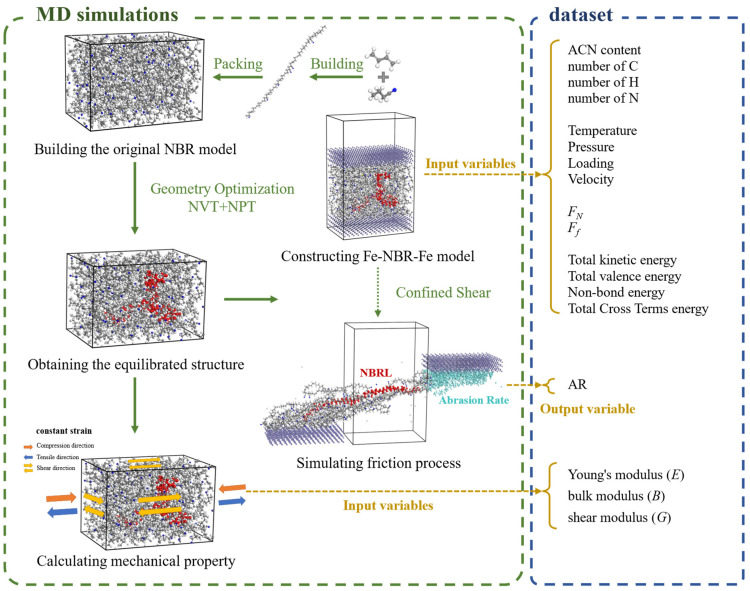
Flow chart of the MD simulations and dataset.

**Figure 2 polymers-15-03799-f002:**
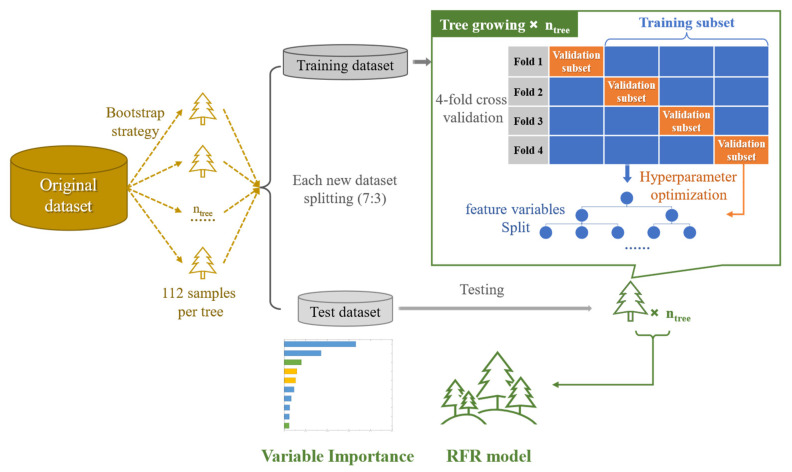
The RFR model. The yellow part denotes the process of new datasets for each tree established by the original dataset. The green box is the growth process of one tree.

**Figure 3 polymers-15-03799-f003:**
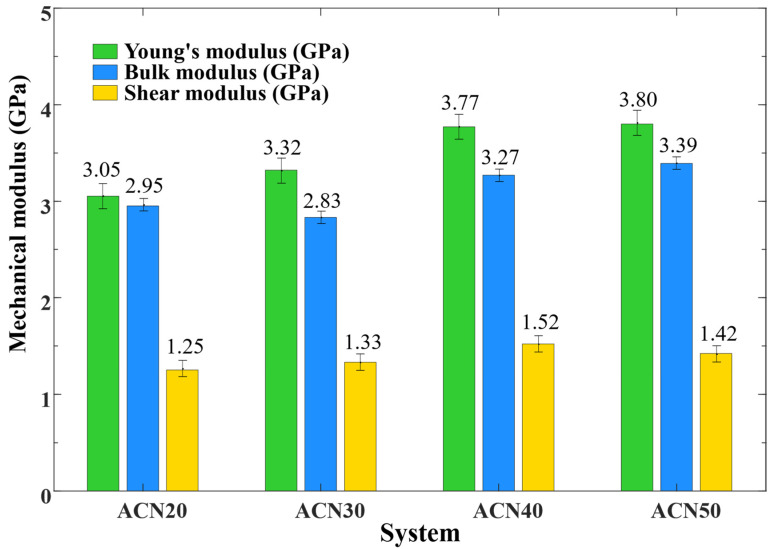
The mechanical properties of NBR.

**Figure 4 polymers-15-03799-f004:**
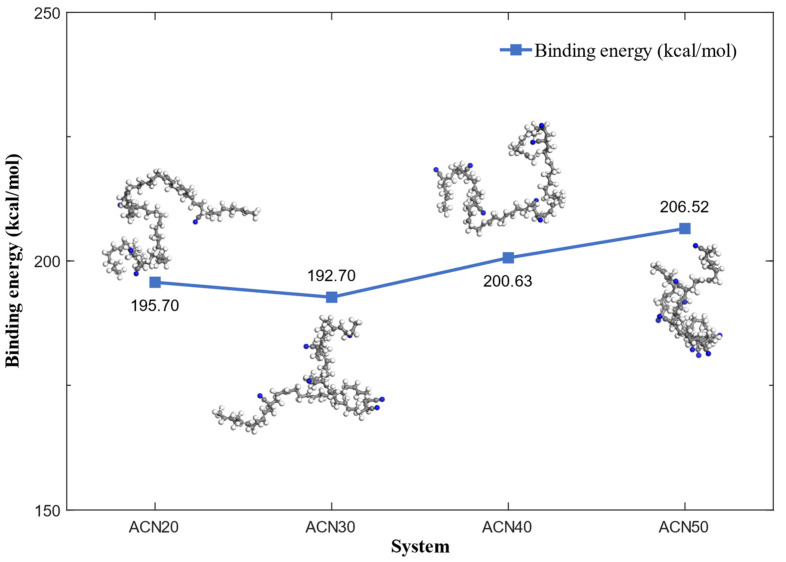
The binding energy between the NBRL and the NBR matrix during the NPT equilibrium processes, and the structure of the NBRL chain.

**Figure 5 polymers-15-03799-f005:**
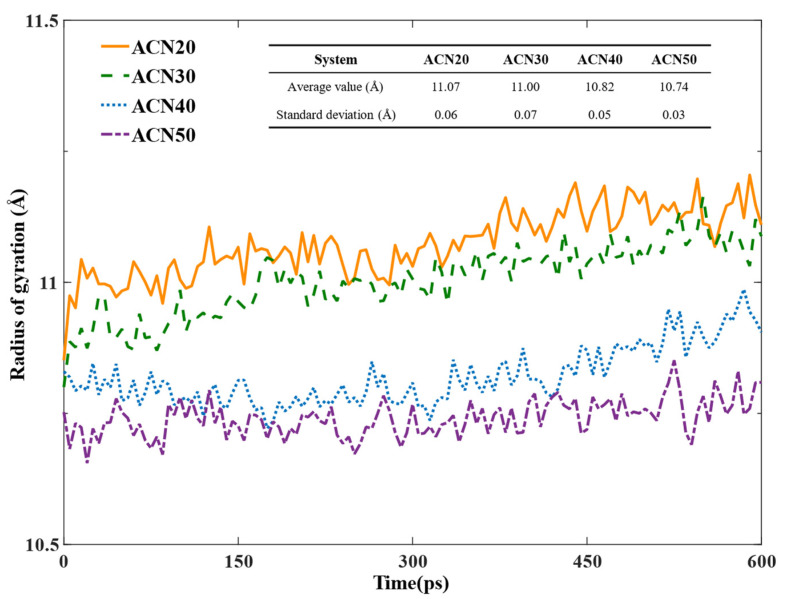
The radius of gyration of NBR.

**Figure 6 polymers-15-03799-f006:**
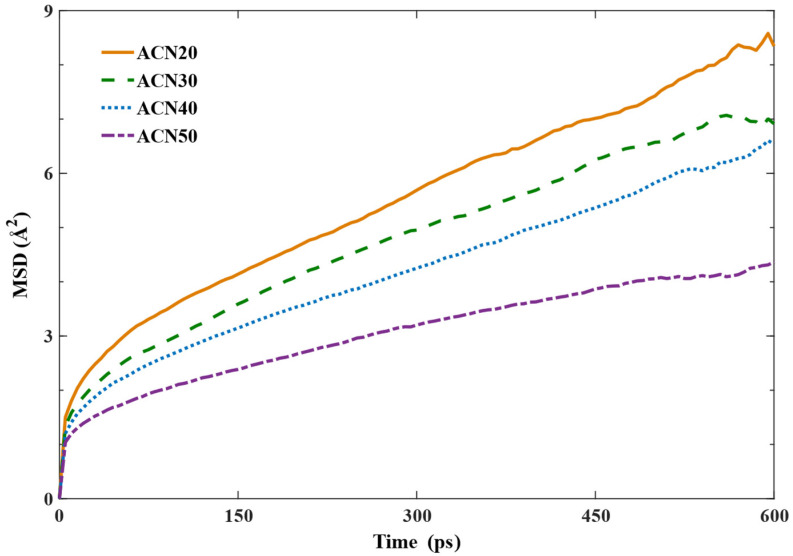
The mean square displacement of NBR.

**Figure 7 polymers-15-03799-f007:**
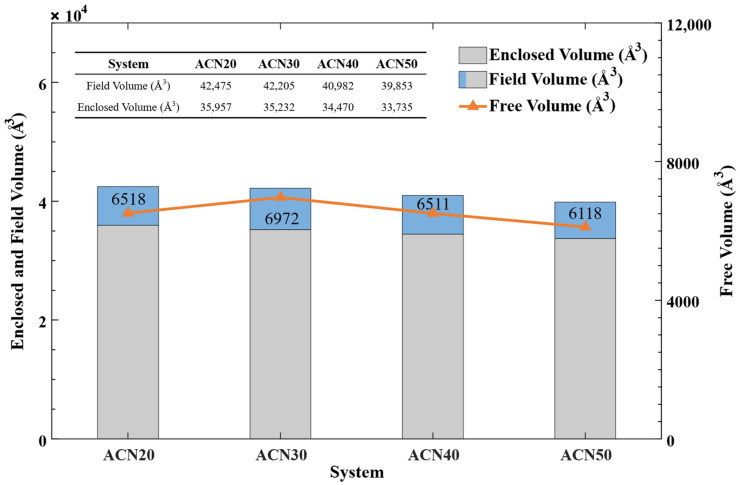
Free volume of NBR.

**Figure 8 polymers-15-03799-f008:**
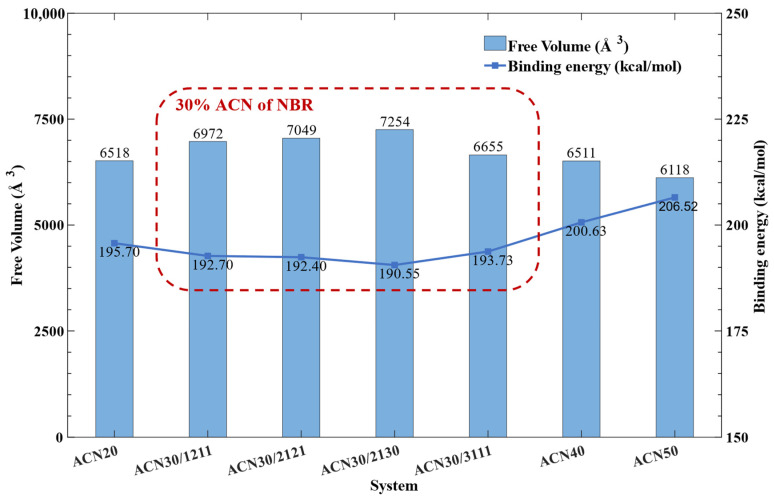
The results of 30% ACN about other NBR models.

**Figure 9 polymers-15-03799-f009:**
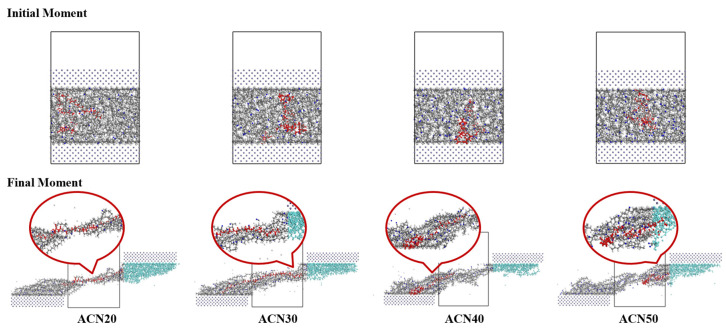
Snapshots of friction processes of NBR. The red chain from the NBR matrix is the NBRL chain. The blue NBR matrix represent the wear atoms during the friction process.

**Figure 10 polymers-15-03799-f010:**
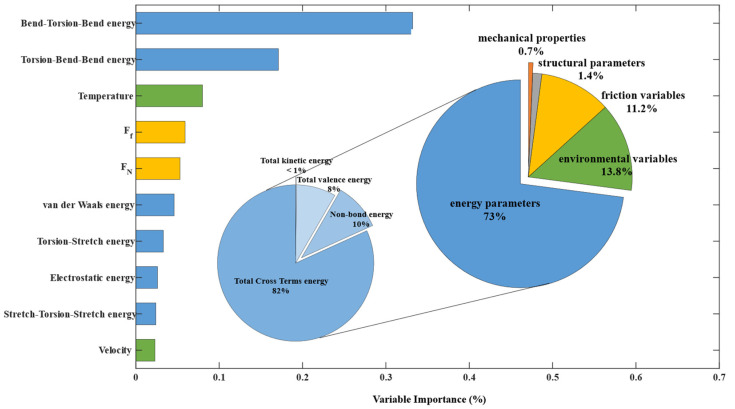
The rankings and proportions of VI for the feature parameters of the RFR.

**Figure 11 polymers-15-03799-f011:**
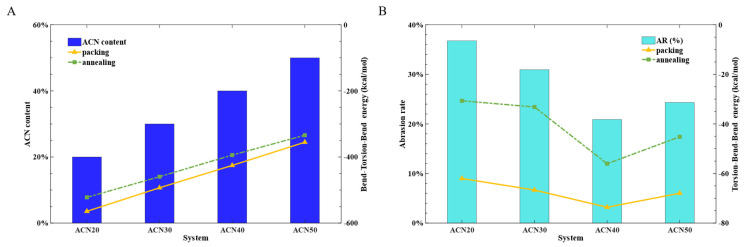
(**A**) The BTB energy versus the ACN content of NBR models. (**B**) The TBB energy versus the AR of NBR models.

**Figure 12 polymers-15-03799-f012:**
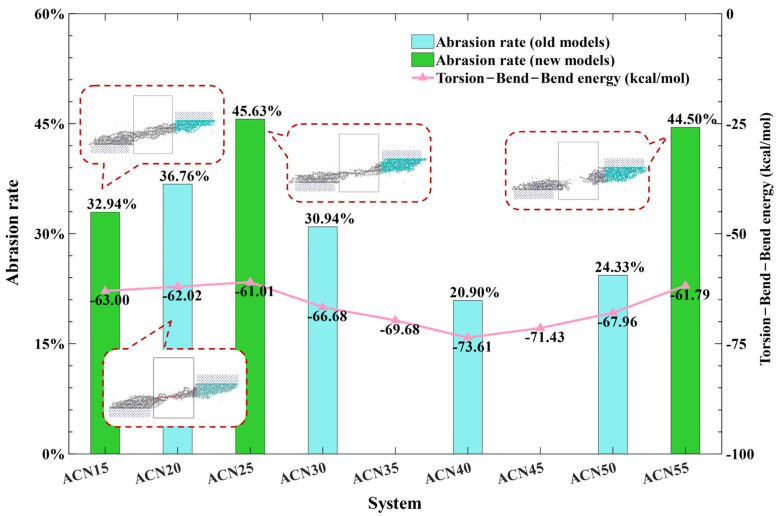
The TBB energy and the AR with all forecasting models.

**Table 1 polymers-15-03799-t001:** Information about the NBR models.

Item	Acrylonitrile and Butadiene Allocation Ratio	ACN Content	Chain Length (Å)	Number of Elements in Single Chain				Number of Total Atoms
				C	H	N	Double Bond	
ACN20	1:4	19.70%	84.33	76	110	4	16	4180
ACN30	3:7	29.61%	80.96	74	104	6	14	4048
ACN40	2:3	39.55%	75.66	72	98	8	12	3916
ACN50	1:1	49.53%	71.74	70	92	10	10	3784

**Table 2 polymers-15-03799-t002:** The dataset of the feature parameters.

Feature Module	The Name of Feature Parameters			
Structural parameters	ACN Content	Number of C	Number of H	Number of N
Mechanical properties	*E*	*G*	*B*	
Friction parameters	F_N_	F_f_	AR	
Environmental variables	Temperature	Pressure	Loading	Velocity
Energy parameters	(Hamiltonian)	(Total valence energy)	(Non-bond energy)	(Total cross terms energy)
	Total kinetic energy	Bond energy	Van der Waals energy	Bend–Bend energy
	(Total potential energy)	Angle energy	Electrostatic energy	Bend–Torsion–Bend energy
		Torsion energy		Torsion–Bend–Bend energy
		Inversion energy		Torsion–Stretch energy
				Stretch–Bend–Stretch energy
				Stretch–Torsion–Stretch energy
				Stretch–Stretch energy

**Table 3 polymers-15-03799-t003:** The mechanical properties of NBR.

System	Young’s Modulus (GPa)	Increase Percentage (%)	Bulk Modulus (GPa)	Increase Percentage (%)	Shear Modulus (GPa)	Increase Percentage (%)
ACN20	3.05	0.00	2.95	0.00	1.25	0.00
ACN30	3.32	8.85	2.83	−4.07	1.33	6.40
ACN40	3.77	23.61	3.27	10.85	1.52	21.60
ACN50	3.80	24.59	3.39	14.92	1.42	13.60

**Table 4 polymers-15-03799-t004:** The information and results of 30% ACN of NBR models with the other arrangement structures of the monomers.

ACN30 Item	The Number of ACN Consecutive Units			Binding Energy (kcal/mol)	Free Volume (Å^3^)
	1 ACN	2 ACN	3 ACN		
1211 (ACN30)	4	1	0	192.70	6972
2121	2	2	0	192.40	7049
2130	1	1	1	190.55	7254
3111	3	0	1	193.73	6655

**Table 5 polymers-15-03799-t005:** The VI ranking and values of RFR model results.

The VI Ranking	RFR Model 1	RFR Model 2	RFR Model 3	RFR Model 4	RFR Model 5	RFR Model 6	RFR Model 7	RFR Model 8
1	BTB energy (62.6%)	BTB energy (33.2%)	BTB energy (41.9%)	BTB energy (26.7%)	BTB energy (37.2%)	BTB energy (33.1%)	BTB energy (51.9%)	BTB energy (41.9%)
2	F_N_ (7.5%)	TBB energy (17.1%)	F_N_ (7%)	TBB energy (13.2%)	F_f_ (9.1%)	F_f_ (10.5%)	TS energy (6.9%)	F_N_ (7%)
3	Temperature (6.1%)	Temperature (8%)	TBB energy (6.8%)	F_N_ (11.9%)	Temperature (8.1%)	TBB energy (10.1%)	F_f_ (6.3%)	TBB energy (6.8%)
4	TBB energy (3.1%)	F_f_ (5.9%)	F_f_ (6.4%)	F_f_ (8.6%)	F_N_ (6.6%)	F_N_ (6.4%)	Temperature (4.9%)	F_f_ (6.4%)
5	F_f_ (2.3%)	F_N_ (5.3%)	STS energy (5.3%)	Temperature (4.4%)	TBB energy (6.5%)	TS energy (5.9%)	F_N_ (4.8%)	STS energy (5.3%)

## Data Availability

Not applicable.

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
