# Peer review of "A Molecular Dynamics Simulation Study on Enhancement of Mechanical and Tribological Properties of Nitrile—Butadiene Rubber with Varied Contents of Acrylonitrile"

_polymers, 2023, doi:10.3390/polym15183799_

Round 1

Reviewer 1 Report (New Reviewer)

In this research, “variable” parameters were built by MD simulations and the applicability of a random forest regression or training was used on an optimization problem of NBR with ACN rubber and its tribologic and mechanical properties.  

Broad comment

The approach to combine MD simulations and random forest regression or training is a very beneficial attempt. The optimization of properties by discussing a large number of feature parameters would be very limited by a classical correlation attempt.

The use and prioritization of variable parameter and their impact on abrasion rate was shortly concluded. A little bit pro contra argumentation of the authors would improve this chapter.

Abtract:

Please reformulate, and restructure the intended meaning: “The snapshots of friction processes of NBR under varied ACN content showed different both the wear forms of NBR matrices and the relative slippage of friction interfaces, owing to the effect of entanglement and interactions  of NBR molecules under the synergy of polarity, rigidity, and chain length”

Please replace by a clear formulation of the findings: “In addition, the variable importance of random forest models screened out the Bend-Torsion-Bend and Torsion-Bend-Bend energy as the highly correlated parameters of ACN and AR, respectively”

Line341: please add a list of the variable parameter, or better introduce variable parameter at the beginning of methods (? ...binding energy, radius of gyration, mean square displacement, and free volume…?)

Line 399 and figure 12: This conclusion is very liberate after the amount of hard work for this paper. “As previously predicting, the AR values of validation models agreed with the trend of the TBB energy, as shown the green bars in Figure 12”. Please add a correlation plot of AR against TBB.

Conclusions

This conclusion should be better prepared in the discussion as stated above for line 399: “The TBB energy as the highly correlated parameter of AR was found by screening VI from RFR model”, or retune the chapter  “Test and verify for identifying result” to “… and correlate for….” or else.

The RFR model and its suitability for conducting the ”regression analysis” should be referred in the conclusion in some manner.

Author Response

We would like to begin with our sincere appreciation for all the valuable comments and insightful suggestions by the reviewers to our manuscript. The comments and suggestions helped us to improve the quality of the manuscript. The methods and the results have been modified. At the same time, the abstract and conclusions are also updated according to the guidance of the reviewers. The modifications indicated below are marked in red in the revised manuscript.

Reviewer 2 Report (New Reviewer)

This work presented a molecular dynamics simulation of the mechanical and tribological properties of nitrile-butadiene rubber (NBR) with acrylonitrile (ACN) enhancement. From their research, they found an optimal ACN content in terms of mechanical properties. They also developed a procedure based on random forest model to identify the important variables correlated to the abrasion rate (AR) of NBR. Overall, this work provides detailed description of the methods they used and sufficient data to support their conclusions. However, I still have some questions and concerns that I hope the authors could answer or clarify.

1.  I suggest the author to increase the font size for Figure 2, especially the upright corner.

2. In Section 3.1, the authors mentioned that there was a turning point at 40% ACN in terms of moduli values. However, from Figure 3, only shear modulus peaks at ACN40. Could the authors make comments on this?

3. I was wondering if the authors could comment on how they calculate the average MSD values.

4. From the discussion at the end of Section 3.1, it seems that ACN30 is more like a turning point. Could the authors comment on this?

5. For random forest model, I understand that it can be used to ‘select’ important variables. However, the description of the work, I was wondering if the authors could just calculate the correlation between the given variable and AR to determine its importance (how well it correlates with AR).

6. Related to the previous question, in addition to finding the important variable, I was wondering if the authors could comment on how they can use the identified variables and random forest model to make predictions of material properties.

1. line 141, 'needs to be'.

2. line 297, 'was meaningful'.

3. line 390, '3.2 Test and verify for identifying result' reads confusing, please check.

4. line 403, 'contributes more to'.

5. line 410, please check the font of 'NBR'.

6. line 425, 'are not strengthened'.

7. line 429, 'the TBB energy'.

Author Response

We thank the reviewer very much for the high evaluation of our work, and for helping us to improve our manuscript. Very professional and insightful comments were obtained from the reviewer to help us improve our manuscript. We appreciate. We studied all the comments from the reviewer carefully and tried our best to improve this manuscript. All the revised places have been marked in red in the revised version. In the followings, we try to make one-to-one replies and discussions according to your questions and concerns from the esteemed reviewer. In addition, we are willing to revise it based on further guidance.

This manuscript is a resubmission of an earlier submission. The following is a list of the peer review reports and author responses from that submission.

Round 1

Reviewer 1 Report

In this study presented by Yuan et al., the influence of ACN content on the mechanical and tribological properties of nitrile-butadiene rubber is studied using MD simulation. The authors further developed an ML model based on random forest regression to predict properties such as abrasion ratio. While the topic is of practical importance, I find the paper too lengthy in discussing trivial topics, and the machine learning results are not as promising as the authors describe in the manuscript. Overall, I think significant improvement of the machine learning experimental design is needed before the manuscript can be considered for publication. Please find my detailed comments below:

1) "Actually, according to the COMPASS force field [18], the total potential energy also includes the total valence and cross-terms energy. These energies could be further subdivided into specific parameters, such as the Bond energy, the Angle energy, the Van der Waals energy, the Bend-Torsion-Bend energy, etc. They were often neglected due to the large quantity of the parameters and little contribution of the value." 

I'm very confused by this claim. By design, these energy terms should be almost universal in commonly used force fields like AMBER / CHARMM, etc. What does the author mean by "little contribution of the value" here?

2) There are only four variations of ACN concentrations studied here (from 20%-50%). Can the authors comment if their conclusions can be interpolated/extrapolated to different ACN ratios?

3) As the authors reveal in Figure 3 (and Figure 5), the ACN30 model seems to show a different trend compared with other systems due to structural effects. Can the authors comment if their results of binding energy/volume, etc., are statistically meaningful (with respect to the structural configurations)?

4) What concerns me most is the design of their ML model. The features selected may be problematic. Firstly, the structural parameters (only the content of monomer and number of atoms) are way too simple. The description in Table 2 is misleading: why does AR, a quantity to be predicted, exist in the feature variables?

5) The use of energy parameters in the ML model seems impractical. This is because these energy terms must be obtained after running an MD annealing. If I understand correctly, with the current model setup and sliding speed, the shearing simulation may even take less time than the initial MD annealing (4.5 nm simulation box and 0.1 Å/ps * 2 --> ~9 ps). 

6) The performance of the RFR model shown in Table 3 needs more explanation. Without showing the detailed dispersion of predicted data, an R^2 of only 0.5 may mean the data are only weakly captured by the model, not meriting "Hence, the reliability and effectiveness were possessed for the developed RFR model in the present work." The authors should show more detail about their RFR predictions and analysis, which is supposedly the main focus of this manuscript.

7) The unit in Table 3 is also questionable. By the definition of AR (Mleave/Mtotal), it should be a value between 0 - 1.0, but all the errors shown are larger than 1. Are the authors actually referring to percentage (AR * 100) during training?

8) The results in Figure 10 may actually be important for this study. From my understanding, performing the additional shear experiments on ACN35 / ACN 45, etc., should be feasible if the equilibrium of these systems is already obtained.

Many of the sentences can be furture improved with the assistance of a native speaker, such as "The TBB energy was finely found to be highly associated with the AR and successfully predicted the AR trend of NBR from new models." in the conclusion part. But in general the manuscript is still readable.

Reviewer 2 Report

Dear,

This was a well-explained piece of work, and I think the simulations shown do lead to some reasonable conclusions. Some details should be clarified more clearly:

> In the introduction, the authors should make clear the motivation for NBR rubber. Why not test on higher-consumption thermoplastics? In addition, the authors must report the limitations of the applied method;

> Page 2. Line 97-104. Did the authors not make it clear how the criteria were adopted? Was it random? In addition, in Table 1, was it also random? Or preliminary studies to adopt the experimental results?

> Figure 2. Are the reported values for NBR correct? Why does this rubber have a high elastic modulus? The level of Polystyrene, Nylon 6, PET, etc.

> Are reported property values an average of many simulations? How many? Please indicate the variation in these results run-to-run, perhaps an error bar or standard deviation.

> Why didn't the authors do the experimental for a comparative effect with the simulation?

> Were the initial properties reported from preliminary results for comparative effect with the simulation?

Minor editing of English language required.

Reviewer 3 Report

While the topic tackled is interesting and falls within the scope of Polymers, the readability and clarity of the manuscript are both extremely poor. Unless significantly re-written I cannot recommend publication. As it is now the manuscript reads more like an internal report than a scientific publication.

) Authors should re-phrase most of the sentences in the manuscript as it is extremely difficult to understand the meaning of many paragraphs including the description of the methodology. If necessary authors should seek help from a native speaker of the English language.

) I could not follow the methodology including the initial procedure for the generation of the structures and the equilibration using MD through the Materials Studio software. It is not clear how the size of the molecules is calculated, neither from which simulations is extracted. In the end it is not clear from which modeling step the data in Table 1 are extracted.

) Simulations should be reproducible from independent researchers but information is missing on the values of pressure and density(volume) in the corresponding ensembles, the duration of simulations etc.

) The energetic terms in Eq. 1 are neither introduced nor explained (some of them are trivial but still some require guess like the correspondence of “oop” to “out of plane”).

) The presented MSD curves correspond to which species? Also, their range suggest that the atoms(?) have moved distances not significantly different than their sizes. Are the corresponding curves thus trustworthy? How long it takes (in real time) to conduct such a simulation? Could it be extended to significantly longer times?

) Which are the definitions of the various types of volumes that are presented in Figure 5? How they are calculated?

)The description in the legends of some tables and figures is telegraphic rending very difficult to understand what is presented. Some are even misleading like in fig. 5 and 6 which suggest free volume (but there are more sets) and density (while volume also appears)

) Snapshot are not very clear as they are rather small. Could they be increased? (like for example the ones shown in Fig. 7).

Whole segments of the manuscript are very unclear and require rephrasing:

Abstract should be rewritten to avoid too many abbreviations and have a clearer message on the computational novelties brought by the present study.

It is not standard in literature to write MDS (molecular dynamics simulations) instead of the most common MD (molecular dynamics) simulations

All abbreviations should be defined upon their introduction (for example CNT etc).

Line 27: “To study…. These models”; Line 40: “polymer molecular” -> “polymer molecules” / “working conditions” -> “process conditions”; Line 47, word “properties” is repeated 5 times in the same sentence; Paragraph in lines 55-66 should be rephrased.

I have commented on this in my review. The manuscript is very unclear and almost inaccessible due to poor syntax and grammar.

Round 2

Reviewer 1 Report

I have read through the revision v1 from the authors for their manuscript "Molecular Dynamics Simulation Combined with Machine Learning for the Tribological Properties of Polymers: an Analysis and Prediction Method". While I find the revisions help address some of my previous concerns, there are several points not so adequately addressed, please find them below:

Comment 3)

The response does not fully address the concern. While a molecular dynamics (MD) simulation can statistically represent the energetic ensemble of a particular system, it does not necessarily ensure that these results are valid for other initial polymer configurations. The notation ACN30 indicates that the polymer contains 30% ACN monomers randomly located in its chain, and there are multiple ways in which the monomers could be arranged. Starting from a different polymer chain configuration for ACN30 alone could generate slightly different values in Figure 3. This is a similar concern to that raised in point 4.

Comment 4)

The reasoning behind "In our RFR algorithms, the dataset in the RFR model needed to include both input and output variables, so AR was also listed as feature variables in Table 2" is unclear. The manuscript should clearly state which variables are input and output variables for the regression model to avoid confusion.

Comment 6)

This is my main concern regarding the manuscript. A parity plot comparing the ground truth and prediction values for individual data points in the validation and test sets would help to verify the performance of the regression model.

Comment 8)

I would encourage the authors to verify their results by trying at least one more system (such as ACN55, as it is an extrapolated data point).

Additional comment

Now that I understand the authors are using the TBB energies directly after geometric optimization as an input variable, the methodology of the regression model seems more practical. However, I am curious about how much variation in the TBB energy from the locally optimized geometry affects the model's prediction results. For example, different starting random seeds when generating the packing geometries may cause the initial geometry to vary and result in different local minima.

Reviewer 3 Report

I cannot recommend publication of the present manuscript to Polymers. While the syntax and grammar have been improved, increasing the clarity of the manuscript the work remains severely flawed. After the authors reconsider their approach the manuscript could be publishable in a more specialized journal.

) MSD curves correspond to very short simulations. As I have commented in my original review the atoms do not displace even distances compared to their dimensions. I would be very surprised if the chain as whole as moved even a fraction. No dynamical information can be extracted from such MD simulations.

Authors in their response about the dynamics cite experimental works instead of simulation-based ones. “Real” times are by no means accessible through all-atom simulations of polymer systems.

) Authors correlate the binding energy in Fig. 3 with the change in conformation of the molecules and the corresponding MSD curve. I do not see any quantification of such dependence. First, it is not clear what exactly is shown in the polymer configurations in Fig. 3. The “red” chain is selected randomly? Does it correspond to the average size of the sample? How, it is possible that a change of 10% content affects so drastically the chain size? Especially when not even the atoms -on average- have been displaced from their original positions?

Then, the clear plateau of Fig. 2 is attributed to the rigidity of the NBR. What is meant by this? How is rigidity quantified here? Entanglements are then mentioned without any further analysis. How rigidity, entanglements and mobility are inter-connected is completely unclear. Some on their quantification.

) Have the authors repeated the simulations by conducting independent ones and calculate the differences? I have very serious concerns on the MD part of the work and its reliability. This is also very relevant to the free volume calculations.

) Authors relate the maximum free volume with the centroid of the molecular spacing, again how the centroid is calculated here? Also, the important change in the density as observed in Fig. 6 for the 50% system is not reflected in the previous results presented. How is this possible?

) Density in y-axis (Fig. 6) is missing units.

Volumes are presented with an accuracy of 0.01A. How is this possible?

The manuscript has been improved in this respect, although some sentences appear and sound strange.